# Differentially Private Multi-Sampling from Distributions

**Albert Cheu**                                                    CHEU@GOOGLE.COM
*Google*

**Debanuj Nayak**                                                 DNAYAK@BU.EDU
*Boston Univeristy*

**Editors:** Gautam Kamath and Po-Ling Loh

## Abstract

Many algorithms have been developed to estimate probability distributions subject to differential privacy (DP): such an algorithm takes as input independent samples from a distribution and estimates the density function in a way that is insensitive to any one sample. A recent line of work, initiated by Raskhodnikova et al. (Neurips '21), explores a weaker objective: a differentially private algorithm that approximates a single sample from the distribution. Raskhodnikova et al. studied the sample complexity of DP *single-sampling* i.e., the minimum number of samples needed to perform this task. They showed that the sample complexity of DP single-sampling is less than the sample complexity of DP learning for certain distribution classes. We define two variants of *multi-sampling*, where the goal is to privately approximate $m > 1$ samples. This better models the realistic scenario where synthetic data is needed for exploratory data analysis.

A baseline solution to *multi-sampling* is to invoke a single-sampling algorithm $m$ times on independently drawn datasets of samples. When the data comes from a finite domain, we improve over the baseline by a factor of $m$ in the sample complexity. When the data comes from a Gaussian, Ghazi et al. (Neurips '23) show that *single-sampling* can be performed under approximate differential privacy; we show it is possible to *single- and multi-sample Gaussians with known covariance subject to pure DP*. Our solution uses a variant of the Laplace mechanism that is of independent interest.

We also give sample complexity lower bounds, one for strong multi-sampling of finite distributions and another for weak multi-sampling of bounded-covariance Gaussians.

**Keywords:** differential privacy, sample complexity, synthetic data

## 1. Introduction

Much research effort has been focused on estimating the parameters of data distributions under the differential privacy (DP) constraint (Dwork et al., 2006). DP algorithms assure *any outlier* that choosing to contribute their data does not greatly increase risk of privacy harms. This is powerful given that the data can concern sensitive personal attributes such as medical history, location history, and internet habits. If we assume data are independent samples from a distribution $\mathbf{D}$, it is natural to estimate the expectation; Karwa and Vadhan (2018) give DP confidence intervals for the Gaussian mean while Kamath et al. (2020) estimate the mean of heavy tailed distributions. Other algorithms perform the more ambitious task of estimating the distribution's density function itself; for example, Kamath et al. (2019) and Bie et al. (2022) show how to estimate Gaussians accurately in statistical distance.

But we do not always require private estimates of the distribution or its parameters. Exploratory data analysis, for example, only requires synthetic samples that in some sense resemble the original distribution. To rigorously define what it means to make such a summary, this work builds on the work of Raskhodnikova et al. (2021). Under their definition, a DP sampling algorithm ingests

independent samples from a distribution $\mathbf{D}$ like prior work but it only emits one random variable that is meant to approximate one fresh sample from $\mathbf{D}$ up to statistical distance $\alpha$. This definition is also adopted by Ghazi et al. (2023), who provide approximate DP algorithms for sampling from Gaussians. We emphasize that each algorithm by these two sets of authors only produces an approximation of a single sample, so we name their objective *single-sampling*. The main motivation of our work is broadening the objective to generating multiple samples under DP.

We note that multiple formalizations of the DP objective exist. Pure DP algorithms, for example, bound the participation risk in terms of a single parameter $\varepsilon$. Meanwhile, approximate DP algorithms permit a $\delta > 0$ chance that the bound does not hold. Other variants of DP like zero concentrated DP (zCDP) interpolate between those two extremes. Our algorithms span all three DP variants.

## 1.1. Our Results and Techniques

Our work extends the prior work along a number of directions.

1. **Novel Definitions.** We formalize what it means to produce $m > 1$ samples from a distribution subject to DP in Section. Specifically, we provide strong and weak variants of *multi-sampling* in Definitions 6 and 7. Both variants require that the output random variables are mutually independent and identically distributed. The weak variant only requires that the marginal distribution of the output is close to that of the input, while the strong variant requires that the product distributions are close.

2. **New Pure DP Single-samplers.** For distributions over the set $[k]$, we show how to perform pure DP single-sampling with a sample complexity that has a smaller leading constant than the prior work by Raskhodnikova et al. (2021) in Theorem 12. We achieve this via amplification-by-subsampling. For multivariate Gaussian distributions with known covariance, we provide a pure DP single-sampler that builds upon the approximate DP single-sampler from Ghazi et al. (2023) in Theorems 19 and 20. We achieve this with a novel variant of the Laplace distribution, which is of independent interest.

3. **Multi-samplers.** We describe baseline techniques to create multi-sampling algorithms from ones designed for single-sampling (see Lemmas 8 & 9 and Corollary 10). In particular, if we apply Corollary 10 to our single samplers, we obtain strong multi-samplers under pure DP for $k$-ary and Gaussian distributions. The sample complexities grow by a factor of $m^2$: one factor of $m$ comes from repeatedly calling the single-sampler, another from a union bound. But under approx. DP we show that one factor of $m$ is sufficient. In the $k$-ary case, amplification-by-shuffling lets us avoid paying the cost of repetitions. In the Gaussian case, Ghazi et al. (2023)'s single-sampler has a sample complexity depending only logarithmically on $\alpha$, mitigating the union bound's effect.

4. **Lower Bounds for Multi-Sampling.** For strong multi-sampling, we formalize the following intuition in Theorem 23: if there are $m$ i.i.d. samples from $\mathbf{D}$ and $m$ i.i.d. samples from $\hat{\mathbf{D}}$ such that the joint random variable is close (i.e., we have a strong multi-sampler) then the individual variables are close (i.e., we have single-sampler). For weak multi-sampling, we note that we can treat the $m$ outputs of a DP sampler as if they were fresh samples *without privacy constraints*. We formalize the intuition that overly-large $m$ would circumvent DP estimation lower bounds, in particular those for Gaussians as shown in Theorem 29.

We summarize our results in Tables 1, 2, and 3. The $\tilde{O}, \tilde{\Omega}, \tilde{\Theta}$ notation suppresses terms that are polylogarithmic in $m, 1/\varepsilon, 1/\delta, 1/\alpha, k, d$.

| $k$-ary distributions | | |
|---|---|---|
| | pure DP | approximate DP |
| Single-Sampling | $\leq \frac{2k}{\alpha\varepsilon}$ 
 Raskhodnikova et al. (2021) 
 $\leq \frac{k}{\alpha\varepsilon}$ 
 Thm. 12 | $\Omega(\frac{k}{\alpha\varepsilon})^*$ 
 Raskhodnikova et al. (2021) |
| Weak Multi-Sampling | $\leq m \cdot \frac{k}{\alpha\varepsilon}$ 
 via Lemma 8 | $O\left(m + \frac{k}{\alpha\min(\varepsilon,\varepsilon^2)}\log\frac{1}{\delta}\right)$ 
 Thm. 13 |
| Strong Multi-Sampling | $\leq m^2 \cdot \frac{k}{\alpha\varepsilon}$ 
 via Coro. 10 | $O\left(m \cdot \frac{k}{\alpha\min(\varepsilon,\varepsilon^2)}\log\frac{1}{\delta}\right)$ 
 Thm. 15 
 $\Omega\left(\sqrt{m}\cdot\frac{k}{\alpha\varepsilon}\right)^*$ 
 Thm 25 |

Table 1: Sample complexity bounds for DP sampling $k$-ary distributions. For the lower bounds $*$, the result assumes $\varepsilon < 1$, $\delta < 1/5000n$, and sufficiently small error $\alpha$.

| Gaussians with known covariance $\Sigma$ | | | |
|---|---|---|---|
| | pure DP | zCDP | approximate DP |
| Single-Sampling | $O\left(\frac{d^{3/2}}{\alpha\varepsilon}\log\frac{d}{\alpha}\right)$ 
 Thm. 20 | $\tilde{O}\left(\frac{\sqrt{d}}{\varepsilon}\right)$ 
 Coro. 22 | $\tilde{\Theta}\left(\frac{\sqrt{d}}{\varepsilon}\right)$ 
 Ghazi et al. (2023) |
| Weak Multi-Sampling via Lemma 8 | $O\left(m \cdot \frac{d^{3/2}}{\alpha\varepsilon}\log\frac{d}{\alpha}\right)$ | $\tilde{O}\left(m \cdot \frac{\sqrt{d}}{\varepsilon}\right)$ | $\tilde{O}\left(m \cdot \frac{\sqrt{d}}{\varepsilon}\right)$ |
| Strong Multi-Sampling via Coro. 10 | $O\left(m^2 \cdot \frac{d^{3/2}}{\alpha\varepsilon}\log\frac{d}{\alpha}\right)$ | $\tilde{O}\left(m \cdot \frac{\sqrt{d}}{\varepsilon}\right)$ | $\tilde{O}\left(m \cdot \frac{\sqrt{d}}{\varepsilon}\right)$ |

Table 2: Sample complexity bounds for DP sampling Gaussians with known covariance matrices.

## 1.2. Open Questions

Our work is an initial foray into DP multi-sampling. As such, there are some questions which we do not have the answers to:

- Under approximate DP, we are able to avoid a multiplicative factor of $m$ in the sample complexity of strong multi-sampling from $k$-ary distributions. But is approximate DP necessary?

| Gaussians with bounded covariance ($I \preceq \Sigma \preceq \kappa I$) | | |
|---|---|---|
| | zCDP | approximate DP |
| Single Sampling | $\tilde{O}_\kappa \left( \frac{d^{3/2}}{\alpha \varepsilon^2} \right)$ 
 Coro. 34 | $\tilde{\Theta}_\kappa \left( \frac{d}{\varepsilon} \right)$ 
 Ghazi et al. (2023) |
| Weak Multi-Sampling via Lemma 8 | $\tilde{O}_\kappa \left( m \cdot \frac{d^{3/2}}{\alpha \varepsilon^2} \right)$ | $\tilde{O}_\kappa \left( m \cdot \frac{d}{\varepsilon} \right)$ |
| Weak Multi-Sampling when $m > d$ | $\tilde{\Omega} \left( \frac{d^2}{\alpha^2} + \frac{d^2}{\alpha \varepsilon} + \frac{1}{\varepsilon} \sqrt{\log \kappa} \right)$ 
 Thm. 29 | |
| Strong Multi-Sampling via Coro. 10 | $\tilde{O}_\kappa \left( m^2 \cdot \frac{d^{3/2}}{\alpha \varepsilon^2} \right)$ | $\tilde{O}_\kappa \left( m \cdot \frac{d}{\varepsilon} \right)$ |

Table 3: Sample complexity bounds for DP sampling Gaussians with bounded covariance matrices.

- Is it possible to perform weak multi-sampling of Gaussians without a multiplicative factor of $m$, as with $k$-ary distributions? Our use of amplification by shuffling is limited, as the known results for local additive noise are weaker than the one for randomized response.

- Perhaps most interestingly, we can conceive of an alternative definition of weak multi-sampling: we could design an algorithm that bounds the distance between the joint distribution of its output and $m$ i.i.d. samples from $\mathbf{D}$, but makes no promises about independence of the members of its output. We call this *qualitatively* weak multi-sampling, as opposed to our *quantitatively* weak multi-sampling.[1] Any strong multi-sampler, like the ones in our work, satisfy both definitions of weak multi-sampling. But what do algorithms tailor-made for qualitatively weak multi-sampling look like? What is the best way to use the ability to correlate random variables? What would their sample complexity be, especially compared to the ones we made for quantitatively weak multi-sampling?

## 2. Preliminaries

### 2.1. Measuring Closeness of Distributions

We briefly review the various ways we will measure how far or close distributions are from each other. Note that we mildly abuse notation and define distances and divergences for distributions and random variables interchangeably.

---

1. The term "quantitative" suits our notion of weak multi-sampling because a weak multi-sampler can be turned into strong multi-sampler by appropriately tuning the error parameter $\alpha$ (Lemma 9). In contrast, independent-and-identically-distributed is a quality that cannot be achieved by setting parameters.

We will measure the error of our sampling algorithms according to *total variation (TV) distance*, also known as *statistical distance*. The TV distance between a pair of distributions $\mathbf{D}, \mathbf{D}'$ is

$$||\mathbf{D} - \mathbf{D}'||_{TV} := \sup_E \left| \mathbb{P}_{\eta \sim \mathbf{D}} [\eta \in E] - \mathbb{P}_{\eta \sim \mathbf{D}'} [\eta \in E] \right|$$

Some variants of differential privacy are phrased in terms of the *Rényi divergence of order* $\alpha$:

$$D_\alpha(\mathbf{D}||\mathbf{D}') := \frac{1}{\alpha - 1} \log \mathbb{E}_{x \sim \mathbf{D}} \left[ \left( \frac{\mathbf{D}(x)}{\mathbf{D}'(x)} \right)^{\alpha - 1} \right]$$

where $\mathbf{D}(x), \mathbf{D}'(x)$ are the densities placed on $x$.

Other versions rely on the *hockey-stick divergence of order* $\beta$:

$$D_\beta^{\mathrm{HS}}(\mathbf{D}||\mathbf{D}') := \sup_E \left( \mathbb{P}_{\eta \sim \mathbf{D}} [\eta \in E] - \beta \cdot \mathbb{P}_{\eta \sim \mathbf{D}'} [\eta \in E] \right)$$

We say that two distributions $\mathbf{D}, \mathbf{D}'$ are $(\varepsilon, \delta)$-*close*—denoted $\mathbf{D} \approx_{\varepsilon, \delta} \mathbf{D}'$ when both $D_{e^\varepsilon}^{\mathrm{HS}}(\mathbf{D}||\mathbf{D}')$ and $D_{e^\varepsilon}^{\mathrm{HS}}(\mathbf{D}'||\mathbf{D})$ are at most $\delta$. When $\delta = 0$, we write $\mathbf{D} \approx_\varepsilon \mathbf{D}'$.

The following technical lemma connects $(\varepsilon, \delta)$-closeness with total-variation distance:

**Lemma 1** *If* $\mathbf{D} \approx_{\varepsilon, \delta} \mathbf{D}'$, *then* $||\mathbf{D} - \mathbf{D}'||_{TV} \leq \frac{2\delta}{e^\varepsilon + 1} + (e^\varepsilon - 1)$

**Proof** From the textbook by Dwork and Roth (2014, Lemma 3.17), there exists $\hat{\mathbf{D}}, \hat{\mathbf{D}}'$ where (a) $||\hat{\mathbf{D}} - \mathbf{D}||_{TV}$ and $||\hat{\mathbf{D}}' - \mathbf{D}'||_{TV}$ are both at most $\frac{\delta}{e^\varepsilon + 1}$ and (b) $\hat{\mathbf{D}} \approx_{\varepsilon, 0} \hat{\mathbf{D}}'$. We can derive $||\hat{\mathbf{D}} - \hat{\mathbf{D}}'||_{TV} \leq e^\varepsilon - 1$ from (b). The lemma follows by the triangle inequality. ∎

### 2.2. Differential Privacy

Our privacy objective is differential privacy (DP). There are multiple variants but all rely on a neighboring relation: two inputs $X, X'$ each containing data from the same $n$ users are neighbors if they differ on the value contributed by one user. This definition of "neighbor" is sometimes referred to as the *replacement* or *bounded* neighboring relation, to contrast with *add-remove* or *unbounded* variant. In the former, neighboring datasets have the same public size $n$. In the latter, one dataset includes a user's data while the other does not (so $n$ is not public knowledge). We prefer the replacement relation because guarantees made for add-remove neighbors make the implicit assumption that $n$ is not public.

**Definition 2 (Pure DP)** *An algorithm $M$ satisfies $\varepsilon$-pure differential privacy (DP) if, for any neighboring inputs $X, X'$, $M(X) \approx_{\varepsilon, 0} M(X')$.*

**Definition 3 (Zero-Concentrated DP)** *An algorithm $M$ satisfies $\frac{\varepsilon^2}{2}$-zero-concentrated differential privacy (zCDP) if, for any neighboring inputs $X, X'$ and any event $E$, $\forall \alpha > 1 \ D_\alpha(M(X)||M(X')) < \alpha \cdot \frac{\varepsilon^2}{2}$.*

**Definition 4 (Approximate DP)** *An algorithm $M$ satisfies $(\varepsilon, \delta)$-approximate differential privacy if, for any neighboring inputs $X, X'$, $M(X) \approx_{\varepsilon, \delta} M(X')$.*

### 2.2.1. RANDOMIZED RESPONSE

A critical component of our results is *k-ary randomized response with parameter $\varepsilon$* (Warner, 1965; Dwork et al., 2006). This is the algorithm that takes a sensitive value $x \in [k]$ as input and produces a value $y \in [k]$ according to the following probability mass function:

$$\mathsf{RR}_x^\varepsilon(y) := \begin{cases} \frac{\exp(\varepsilon)}{\exp(\varepsilon)+k-1} & \text{if } y = x \\ \frac{1}{\exp(\varepsilon)+k-1} & \text{otherwise} \end{cases} \tag{1}$$

This satisfies $\varepsilon$-DP as every outcome's assigned probability changes multiplicatively by $\exp(\varepsilon)$ or $\exp(-\varepsilon)$.

## 2.3. Definitions of Sampling Tasks

We now define the tasks of interest under DP. Let $\mathcal{D}$ be a family of distributions (e.g. Gaussians or distributions over $[k]$). We will use $n$ to denote sample complexity, $m$ to denote the number of desired samples, and $\alpha$ to denote an error tolerance.

**Definition 5 (Single-Sampling Raskhodnikova et al. (2021))** *An algorithm A performs $\alpha$-sampling for a class $\mathcal{D}$ of distributions with sample complexity $n \in \mathbb{Z}$ if the following holds for any $\mathbf{D} \in \mathcal{D}$: algorithm A consumes $n$ i.i.d. samples from $\mathbf{D}$ in order to produce one sample from some distribution $\hat{\mathbf{D}}$ where $||\mathbf{D} - \hat{\mathbf{D}}||_{TV} \leq \alpha$. The randomness of $\hat{\mathbf{D}}$ comes from both the $n$ samples of $\mathbf{D}$ and the coins of algorithm A.*

**Definition 6 (Strong Multi-sampling)** *An algorithm A performs strong $(m, \alpha)$-sampling for $\mathcal{D}$ with sample complexity $n$ if the following holds for any $\mathbf{D} \in \mathcal{D}$: when A consumes $n$ independent samples from $\mathbf{D}$, it produces $m$ independent samples from some distribution $\hat{\mathbf{D}}$ where $||\mathbf{D}^{\otimes m} - \hat{\mathbf{D}}^{\otimes m}||_{TV} \leq \alpha$.*

**Definition 7 (Weak Multi-sampling)** *An algorithm A performs weak $(m, \alpha)$-sampling for $\mathcal{D}$ with sample complexity $n$ if the following holds for any $\mathbf{D} \in \mathcal{D}$: algorithm A consumes $n$ i.i.d. samples from $\mathbf{D}$ in order to produce $m$ i.i.d. samples from some distribution $\hat{\mathbf{D}}$ where $||\mathbf{D} - \hat{\mathbf{D}}||_{TV} \leq \alpha$.*

We briefly illustrate the difference between strong and weak multi-sampling. Consider an algorithm that consumes $m$ samples from $\mathbf{D}$ and produces some analysis of $\mathbf{D}$ (e.g. quantile). If it were given the output of a strong multi-sampler instead of $m$ samples from $\mathbf{D}$, the analysis will remain $\alpha$-close to its non-private counterpart. Meanwhile, consuming the output of a weak multi-sampler would lead to conclusions about a distribution $\alpha$ close to $\mathbf{D}$.

### 2.3.1. BASELINE TECHNIQUES FOR MULTI-SAMPLING

Here, we describe baseline techniques to perform weak and strong multi-sampling. We show that it is possible to use a single-sampler for weak multi-sampling and that it is possible to use a weak multi-sampler for strong multi-sampling.

**Lemma 8 (From Single to Weak Multi-sampling)** *If A performs $\alpha$-sampling for $\mathcal{D}$ with sample complexity $n$, then there is an algorithm $A'$ that performs weak $(m, \alpha)$-sampling for $\mathcal{D}$ with sample complexity $m \cdot n$. Specifically, $A'$ executes A on disjoint sets of $n$ samples. $A'$ has the same differential privacy guarantee as A.*

**Lemma 9 (From Weak to Strong Multi-sampling)** *If $A$ performs weak $(m, \alpha)$-sampling for $\mathcal{D}$ with sample complexity $n(\alpha)$, then there is an algorithm $A'$ that performs strong $(m, \alpha)$-sampling for $\mathcal{D}$ with sample complexity $n(\alpha/m)$. Specifically, $A'$ executes $A$ with a small enough error tolerance to apply a union bound. $A'$ has the same differential privacy guarantee as $A$.*

Chaining the two transformations together leads to the following corollary:

**Corollary 10 (From Single-Sampling to Strong Multi-sampling)** *If $A$ performs $\alpha$-sampling for $\mathcal{D}$ with sample complexity $n(\alpha)$, then there is an algorithm $A'$ that performs strong $(m, \alpha)$-sampling for $\mathcal{D}$ with sample complexity $m \cdot n(\alpha/m)$. $A'$ has the same differential privacy guarantee as $A$.*

### 2.3.2. FAMILIES OF DISTRIBUTIONS

For any positive integer $k$, the family of $k$-ary distributions consists of all distributions over the set $[k] := \{1, \ldots, k\}$. We will sometimes use "finite-domain distributions" to refer to them.

For any positive integer $d$ and positive reals $\kappa, R$, we use $N^d(\leq R, \preceq \kappa)$ to denote the Gaussian distributions that have mean $\mu \in \mathbb{R}^d$ and covariance matrix $\Sigma \in \mathbb{R}^{d \times d}$ satisfying $\|\mu\|_2 \leq R$ and $I \preceq \Sigma \preceq \kappa \cdot I$. We will call this a family of bounded-mean and bounded-covariance Gaussians. We drop $d$ when it is clear from context.

We note special cases of the above. The symbol $N(\leq \infty, \preceq \kappa)$ (resp. $N(\leq R, \preceq \infty)$) refers to all Gaussians with $\kappa$-bounded covariance but unbounded mean (resp. $R$-bounded mean but unbounded covariance). The symbol $N(\leq R, \Sigma)$ refers to all Gaussians with $R$-bounded mean and covariance matching $\Sigma$.

Ghazi et al. (2023) in their paper show that a result by Ghazi et al. (2020, Theorem 6) implies that DP algorithms that perform sampling for $N^d(\leq R, \preceq \kappa)$ can be turned ones that perform sampling for $N^d(\leq \infty, \preceq \kappa)$.

**Lemma 11 (Reduction for Unbounded Mean)** *For any $d \geq 1, R > 0, \varepsilon > 0, \delta \in [0, 1)$, and $\alpha, \delta' \in (0, 1)$. If there exists an $(\varepsilon, \delta \geq 0)$-DP algorithm for $\alpha$-sampling of $N^d(\leq R, \preceq \kappa)$ with sample complexity $n(d, R, \alpha, \varepsilon, \delta)$, then*

- *(Pure DP reduction). There exists an $(2\varepsilon, \delta)$-DP algorithm for $2\alpha$-sampling of $N^d(\leq \infty, \preceq \kappa)$ with sample complexity $n(d, O(\kappa\sqrt{d}), \alpha, \varepsilon, \delta) + \tilde{O}(d \log(d)/\varepsilon)$.*

- *(Approximate DP reduction). There exists an $(2\varepsilon, \delta + \delta')$-DP algorithm for $2\alpha$-sampling of $N^d(\leq \infty, \preceq \kappa)$ with sample complexity $n(d, O(\kappa\sqrt{d}), \alpha, \varepsilon, \delta) + \tilde{O}(\sqrt{d} \log(d)/\varepsilon)$.*

*The same holds for both variants of $(m, \alpha)$-sampling.*

We also a provide a proof sketch for the above for completeness' sake in Appendix A.

## 3. DP Sampling Algorithms for Finite-Domain Distributions

In this section we describe DP sampling algorithms for $k$-ary distributions. Our analysis is based upon privacy amplification: we develop a local randomizer with a large privacy parameter $\varepsilon_0$, then argue random sampling or shuffling shrinks the effective privacy parameter $\varepsilon$. We formalize the intuition that the large local $\varepsilon_0$ means that the samples are not too polluted by DP noise.

We note that Appendix B will detail extensions when we have hints about the shape of the distribution.

### 3.1. Single-Sampling

As a warm up to multi-sampling, we first show how to perform DP single-sampling by picking a random sample and then performing randomized response. We can express the output distribution as a mixture between the input distribution and the uniform distribution; we bound the weight on the uniform distribution by $\alpha$. The sample complexity bound given by Raskhodnikova et al. (2021) has a leading constant of 2, while our constant is $\leq 1$. Both algorithms are asymptotically optimal, as Raskhodnikova et al. (2021) proved a lower bound of $\Omega(k/\alpha\varepsilon)$.

---

**Algorithm 1:** SubRR, Subsampled Randomized Response

---

**Parameter**: $\varepsilon$
**Input:** $n$ samples $X_1, \ldots X_n$ from a distribution over $[k]$
**Output:** One sample $\hat{X}$ from a distribution over $[k]$
Assign $\varepsilon_0 \leftarrow \ln(\varepsilon \cdot n)$
Choose $r \in [n]$ uniformly at random
Compute $\hat{X}$ by running $k$-ary randomized response on $X_r$ with parameter $\varepsilon_0$
**return** $\hat{X}$

---

**Theorem 12** *For any $\varepsilon > 0$ and $\alpha \in (0, 1)$,* SubRR *is $\varepsilon$-DP and performs $\alpha$-sampling for distributions over $[k]$ with sample complexity* $\frac{1}{\alpha\varepsilon} \cdot (k-1)(1-\alpha) \leq \frac{k}{\alpha\varepsilon}$.

**Proof** We begin with the privacy proof. For any outcome $y \in [k]$ and $X, X'$ that differ on index $j$, the ratio between the likelihood of $y$ is

$$\frac{\mathbb{P}\left[\mathsf{SubRR}(X) = y\right]}{\mathbb{P}\left[\mathsf{SubRR}(X') = y\right]} = \frac{\sum_i \frac{1}{n}\mathsf{RR}^{\varepsilon_0}_{X_i}(y)}{\sum_i \frac{1}{n}\mathsf{RR}^{\varepsilon_0}_{X'_i}(y)} = \frac{\sum_{i \neq j} \frac{1}{n}\mathsf{RR}^{\varepsilon_0}_{X_i}(y)}{\sum_i \frac{1}{n}\mathsf{RR}^{\varepsilon_0}_{X'_i}(y)} + \frac{\frac{1}{n}\mathsf{RR}^{\varepsilon_0}_{X_j}(y)}{\sum_i \frac{1}{n}\mathsf{RR}^{\varepsilon_0}_{X'_i}(y)}. \tag{2}$$

The first step comes from the definition of randomized response.

In the first summand, there is one more term in the denominator's summation than the numerator's summation. The excess term is a probability so it is $> 0$. Moreover, because $X, X'$ differ only on $j$, all the other terms match. Hence

$$(2) < 1 + \frac{\frac{1}{n}\mathsf{RR}^{\varepsilon_0}_{X_j}(y)}{\sum_i \frac{1}{n}\mathsf{RR}^{\varepsilon_0}_{X'_i}(y)} \leq 1 + \frac{\frac{1}{n}\mathsf{RR}^{\varepsilon_0}_{X_j}(y)}{\min_i \mathsf{RR}^{\varepsilon_0}_{X'_i}(y)}$$

To bound the second summand, we recall that randomized response is $\varepsilon_0$-local DP: the above is bounded by $1 + \frac{1}{n}\exp(\varepsilon_0)$ This in turn is bounded by $1 + \varepsilon \leq \exp(\varepsilon)$ by virtue of our choice of $\varepsilon_0$.

We now argue that the output of SubRR, $\hat{X}$, will have TV distance $\alpha$ from $\mathbf{D}$ when run on independent samples from $\mathbf{D}$. Observe that the random variable obtained by executing randomized response on any $x \in [k]$ is distributed as the mixture

$$\frac{\exp(\varepsilon_0)}{\exp(\varepsilon_0) + k - 1} \cdot \mathbf{P}_x + \frac{k-1}{\exp(\varepsilon_0) + k - 1} \cdot \mathbf{U}_{[k]-x}$$

where $\mathbf{P}_x$ denotes the point distribution supported on $x$ and $\mathbf{U}_{[k]-x}$ is the uniform distribution over $[k]$ excluding $x$. When $x$ is drawn from some $\mathbf{D}$, the random variable is distributed as

$$\frac{\exp(\varepsilon_0)}{\exp(\varepsilon_0) + k - 1} \cdot \mathbf{D} + \frac{k-1}{\exp(\varepsilon_0) + k - 1} \cdot \sum_{x \in [k]} \mathbb{P}_{\hat{x} \sim \mathbf{D}}[\hat{x} = x] \cdot \mathbf{U}_{[k]-x}$$

Finally, we show that the mixture weight $\frac{k-1}{k-1+\exp(\varepsilon_0)}$ is at most $\alpha$, which means the output of SubRR follows a distribution $\alpha$-close to $\mathbf{D}$.

$$\frac{k-1}{k-1+\exp(\varepsilon_0)} = \frac{k-1}{k-1+\varepsilon n} \leq \frac{k-1}{k-1+(k-1)(1-\alpha)/\alpha} = \frac{1}{1+(1-\alpha)/\alpha} = \alpha$$

The first step comes from our choice of $\varepsilon_0$ and the second comes from our bound on $n$. ∎

### 3.2. Weak Multi-Sampling

We pivot to the weak multi-sampling problem. A baseline solution is to leverage Lemma 8: repeatedly invoke the single-sampler on disjoint batches of samples. This would cost us a factor of $m$ in the sample complexity. But we show that the overhead can be reduced by simply changing subsampling to shuffling.

---

**Algorithm 2:** ShuRR, Shuffled Randomized Response

---

**Parameters**: $\varepsilon, \delta > 0$ and $f : \mathbb{R}^+ \to \mathbb{R}^+$
**Input:** $n$ samples $X_1, \ldots X_n$ from a distribution over $[k]$
**Output:** $m$ samples $\hat{X}_1, \ldots, \hat{X}_m$ from a distribution over $[k]$ where $m \leq n$
Assign $\varepsilon_0 \leftarrow \ln\left(\frac{f^2(\varepsilon)n}{\ln(4/\delta)} - 1\right)$
Execute randomized response on $X_1, \ldots, X_n$ with parameter $\varepsilon_0$ then shuffle the results
**return** *the first $m$ elements of the shuffled results*

---

**Theorem 13** *There exists a choice of $f : \mathbb{R}^+ \to \mathbb{R}^+$ such that, for any $0 < \varepsilon, \delta, \alpha < 1$ and $m \in \mathbb{N}$, ShuRR is $(\varepsilon, \delta)$-DP and performs weak $(m, \alpha)$-sampling with sample complexity $n = O\left(m + \frac{k}{\alpha\varepsilon^2} \log \frac{1}{\delta}\right)$. There is another $f$ which ensures $\varepsilon$-DP for $\varepsilon > 1$; the sample complexity becomes $O\left(m + \frac{k}{\alpha\varepsilon} \log \frac{1}{\delta}\right)$.*

Note that the ratio between our algorithm's sample complexity and $m$—the average cost to generate each of the $m$ samples—is $O_\delta(1 + \frac{k}{m\alpha\varepsilon})$, a function of $m$ that approaches 1 from above. For comparison the naive approach of repeatedly executing a DP single-sampler has average sample cost $\Omega(k/\alpha\varepsilon)$.

We provide a proof for Theorem 13 in Appendix C, but sketch the ideas here. Just as before, we express randomized response as a mixture distribution to argue that the error parameter is $\leq \alpha$. To argue that the algorithm ensures a target level of DP, we use the amplification-by-shuffling lemma of Feldman et al. (2021) instead of amplification-by-subsampling.

**Remark 14 (Improvements to Analysis)** *The privacy analysis of ShuRR is a black-box invocation of an amplification-by-shuffling lemma. Future work that develops tighter bounds on the amplification phenomenon can be swapped in without disturbing the heart of the argument.*

### 3.3. Strong Multi-Sampling

We conclude with strong multi-sampling. We leverage Lemma 9: if we invoke our weak multi-sampling result with an $\alpha/m$ parameter, we can use a union bound to argue the joint distribution of the $m$ samples is $\alpha$-close to the target.

**Theorem 15** *There exists a choice of* $f : \mathbb{R}^+ \to \mathbb{R}^+$ *such that, for any* $0 < \varepsilon, \delta, \alpha < 1$ *and* $m \in \mathbb{N}$, ShuRR *is* $(\varepsilon, \delta)$-*DP and performs strong* $(m, \alpha)$-*sampling with sample complexity* $n = O\left(\frac{mk}{\alpha\varepsilon^2} \log \frac{1}{\delta}\right)$. *There is another* $f$ *which ensures* $\varepsilon$-*DP for* $\varepsilon > 1$; *the sample complexity becomes* $O\left(\frac{mk}{\alpha\varepsilon} \log \frac{1}{\delta}\right)$.

## 4. DP Sampling Algorithms for Gaussian Distributions

Ghazi et al. (2023) gave the first approximate DP algorithms for single sampling from Gaussians where the covariance is either known, bounded or unknown. Weak and strong multi-sampling algorithms can be derived for all of these cases using Lemmas 8 and 9. We build upon the algorithms of Ghazi et al. to present the first pure DP algorithms for the known covariance case. Furthermore, we demonstrate that one of their algorithms satisfies the stricter privacy notion of zCDP. Additionally, we examine Gaussians with bounded covariance in appendix D.

**Remark 16** *The sample complexity of the approximate DP* $\alpha$-*sampling algorithms from Ghazi et al. (2023, Table 1) only has a logarithmic dependence on* $1/\alpha$, *thus applying Lemma 9 leads to a factor of* $m$ *and not* $m^2$ *in the sample complexity for strong* $(m, \alpha)$ *multi-sampling as seen in Tables 2 and 3.*

### 4.1. Known Covariance, Pure Differential Privacy

In this section, we describe pure DP algorithms for single-sampling from Gaussians with known covariance matrix $\Sigma$. It is without loss of generality to assume it is the identity matrix $I$ since we can apply the transform $\Sigma^{-1/2}$. Weak and strong multi-sampling algorithms can be derived by way of Lemmas 8 and 9. It is an interesting open question whether we can avoid a factor of $m$ as with $k$-ary distributions.

#### 4.1.1. BUILDING-BLOCK: THE EUCLIDEAN-LAPLACE MECHANISM

A core building block of this section is a generalization of the Laplace distribution to $d$-dimensional space (and a corresponding variant of the Laplace mechanism). The generalization that is most common in the DP literature is comprised of one independent scalar Laplace random variable per dimension. It is enough to ensure pure DP for $\ell_1$-sensitive functions. Here, we instead calibrate to $\ell_2$-sensitivity by defining a density function in terms of the Euclidean distance.

Let $\mathbf{ELap}(b)$ denote the distribution over $\mathbb{R}^d$ with one[2] parameter $b > 0$ and density function

$$F_{\mathbf{ELap}(b)}(\eta) = \frac{\Gamma(d/2)}{2\pi^{d/2} b^d \Gamma(d)} \cdot \exp\left(-\frac{\|\eta\|_2}{b}\right) \tag{3}$$

Proving that the above function integrates to 1 is an exercise we defer to Appendix E. Note that the exponential term is a generalization of the exponential term of the scalar Laplace distribution: the absolute value simply became the Euclidean norm. We call this distribution as the *Euclidean-Laplace* distribution. Note that similar generalizations such as the *Generalized Gaussian mechanism* (Ganesh and Zhao, 2021) have been introduced before where the noise is sampled proportional to $\exp\left\{-\left(\frac{\|\eta\|_p}{b}\right)^p\right\}$ for some $p$.

It will be very useful to bound the norm of a Euclidean-Laplace random variable:

---

2. It is natural to define a version with variable mean $\mu$ but we omit that for neatness.

**Lemma 17** *For any $\alpha \in (0,1)$ and $\eta \sim \textbf{ELap}(b)$, $\mathbb{P}\left[\|\eta\|_2 > db \ln \frac{d}{\alpha}\right] \leq \alpha$.*

We defer the proof to Appendix E. In Algorithm 3, we provide pseudocode for the Euclidean-Laplace mechanism for private sums of vectors with bounded norm.

---

**Algorithm 3:** The Euclidean-Laplace mechanism

---

**Input:** $X_1, \ldots, X_n \in \mathbb{R}^d$ that each satisfy $\|X_i\|_2 \leq B$
**Output:** $y \in \mathbb{R}^d$
**Parameters:** $B, \varepsilon > 0$
$b \leftarrow B/\varepsilon$
$\eta \sim \textbf{ELap}(b)$
$y \leftarrow \eta + \sum_{i \in [n]} X_i$
**return** $y$

---

It is straightforward to prove that the Euclidean-Laplace mechanism ensures pure DP.

**Theorem 18** *For any $B, \varepsilon > 0$, the Euclidean-Laplace mechanism satisfies $\varepsilon$-DP.*

**Proof** Fix any sequence of inputs $X, X'$ that differ on exactly one index $i$. Let $S$ (resp. $S'$) be shorthand for $\sum_{i \in [n]} X_i$ (resp. $\sum_{i \in [n]} X_i'$). Then for any event $E \subseteq \mathbb{R}^d$,

$$\frac{\Pr[S + \eta \in E]}{\Pr[S' + \eta \in E]} = \frac{\int_{y \in E} F_{\textbf{ELap}(b)}(y - S)dy}{\int_{y \in E} F_{\textbf{ELap}(b)}(y - S')dy} \leq \max_{y \in E} \frac{F_{\textbf{ELap}(b)}(y - S)}{F_{\textbf{ELap}(b)}(y - S')} \leq e^{\frac{\varepsilon \cdot \|S - S'\|_2}{B}} \leq e^{\varepsilon}.$$

∎

### 4.1.2. SINGLE-SAMPLING, WITH BOUNDED MEAN

Assuming finite $R > 0$, we describe how to perform single-sampling for $N^d(\leq R, I)$ while satisfying pure differential privacy. The pseudocode can be found in Algorithm 4. It is a modified version of the Gaussian mechanism as constructed by Ghazi et al. (2023) for approximate DP sampling.

---

**Algorithm 4:** Our Gaussian Single-Sampler

---

**Input:** $X_1, \ldots, X_n \in \mathbb{R}^d$
**Output:** $y \in \mathbb{R}^d$
**Parameters:** $B, \varepsilon > 0$
$\{\hat{X}_i \leftarrow X_i \cdot \min\left(\frac{B}{\|X_i\|_2}, 1\right)\}_{i \in [n]}$
$y \leftarrow$ the outcome of the Euclidean-Laplace mechanism on $\hat{X}$ with parameters $B, \varepsilon$
$\sigma^2 \leftarrow (n-1)/n$
$Z \sim N(0, \sigma^2 \cdot I)$
$y \leftarrow Z + \frac{1}{n} \cdot y$
**return** $y$

---

**Theorem 19** *There is a constant $c$ such that, when $B \leftarrow R + c \cdot \sqrt{d \log(1/\alpha)}$, Algorithm 4 is $\varepsilon$-DP and performs $O(\alpha)$-sampling for $N^d(\leq R, I)$ with sample complexity $\tilde{O}(\frac{d^{3/2} + dR}{\varepsilon \alpha} \log \frac{d}{\alpha})$.*

**Proof** Let $M$ be the present mechanism. By closure of DP under post-processing and the fact that we ensured all the inputs have norm at most $B$, $M$ is $\varepsilon$-DP. So it simply remains to argue that, when given i.i.d. samples from $N(\mu, I)$, this mechanism's output closely resembles one sample from $N(\mu, I)$.

Consider the alternative mechanism $M'$ where $\eta$ is deterministically set to the 0 vector. Note that $M'$ is found in the prior work by Ghazi et al. (2023) (their Algorithm 1). We write $N(\mu, I)^n$ as shorthand for $n$ independent samples from $N(\mu, I)$. We will we show $M(N(\mu, I)^n) \approx_{\alpha, 2\alpha} M'(N(\mu, I)^n)$, so that we get $||M'(N(\mu, I)^n) - M(N(\mu, I)^n)||_{TV} \leq 6\alpha$ via Lemma 1 and the fact that $\alpha \in (0, 1)$.

Ghazi et al. (2023) show that $||M'(N(\mu, I)^n) - N(\mu, I)||_{TV} \leq \alpha$ in TV-distance (their Theorem 4.2). Via the triangle inequality, we have that the TV distance between $M(N(\mu, I)^n)$ and $N(\mu, I)$ is $7\alpha$.

Thus, it remains to prove $M(N(\mu, I)^n) \approx_{\alpha, 2\alpha} M'(N(\mu, I)^n)$. To do so, we will bound the norm of our Euclidean-Laplace noise vector then invoke the differential privacy offered by Gaussian noise. More precisely, Lemma 17 implies that our sample from the Euclidean-Laplace distribution $\eta$ has Euclidean norm at most $O(dB \log(d/\alpha)/\varepsilon)$ except with probability $\alpha$. Because $Z$ is a Gaussian with covariance $\frac{n-1}{n} \cdot I$, the quantity $Z + \eta/n$ is an instance of the Gaussian mechanism: for our range of $n$ and $\eta$, we argue that $Z + \eta/n \approx_{\alpha, \alpha} Z$. Hence, $M(N(\mu, I)^n) \approx_{\alpha, 2\alpha} M'(N(\mu, I)^n)$.

To prove $Z + \eta/n \approx_{\alpha, \alpha} Z$, we recall that adding Gaussian noise of variance $O(\frac{\Delta_2^2}{\alpha^2} \log(1/\alpha))$ to each coordinate of a $\Delta_2$-sensitive sum suffices to ensure $\alpha, \alpha$-DP (Dwork and Roth, 2014). Here, $\Delta_2$ is the maximum Euclidean norm of $\eta/n$, which we have established is $O(dB \log(d/\alpha)/\varepsilon n)$. The variance in each coordinate of $Z$ is 1, so it suffices for $n$ to be $O(dB \log(d/\alpha) \log(1/\alpha)/\alpha \varepsilon)$. ∎

### 4.1.3. SINGLE-SAMPLING WITH UNBOUNDED MEAN

**Theorem 20** *Let $\alpha \in (0, 1), \varepsilon > 0$. There exists an $\varepsilon$-DP algorithm that performs $\alpha$-sampling for $N^d(\leq \infty, I)$ with sample complexity $\tilde{O}\left(\frac{d^{3/2}}{\alpha \varepsilon} \log \frac{d}{\alpha}\right)$.*

**Proof** Theorem 19 tells us that algorithm 4 is a pure DP algorithm that performs $\alpha$-sampling for $N(\leq R, I)$ with sample complexity $\tilde{O}(\frac{d^{3/2} + dR}{\varepsilon \alpha} \log(d/\alpha))$. Using the pure DP reduction of Lemma 11 with $\kappa = 1$ yields a pure DP algorithm that performs $\alpha$-sampling for $N(\leq \infty, I)$ with sample complexity $\tilde{O}\left(\frac{d^{3/2}}{\varepsilon \alpha} \log \frac{d}{\alpha}\right)$. ∎

### 4.2. Known Covariance, zero-concentrated DP

Ghazi et al. (2023) provide an approximate DP algorithm (their Algorithm 1) for single sampling from $N(\leq R, I)$. In this section, we show that this algorithm also satisfies the stricter privacy notion of $\frac{\varepsilon^2}{2}$-zCDP.

Briefly, this algorithm takes as input samples from $N^d(\leq R, I)$, clips them in a ball of radius $B = R + O(\sqrt{d + \log(1/\alpha)})$, adds Gaussian noise of appropriate variance, $Z \sim N(0, \sigma^2)$ to the empirical mean of the clipped input samples then outputs it.

**Theorem 21 (Bounded Mean)** *There exists an $\frac{\varepsilon^2}{2}$-zCDP algorithm that performs $\alpha$-sampling for $N^d(\leq R, I)$ with sample complexity $\tilde{O}\left(\frac{R+\sqrt{d}}{\varepsilon}\right)$.*

**Proof** We will use the same set of parameters that Ghazi et al. (2023) use, setting $\sigma^2 = \frac{n-1}{n}$ and letting $B = R + O(\sqrt{d + \log(1/\alpha)})$. Note that the accuracy analysis stays the same as there is no change to the algorithm. For the zCDP privacy analysis, notice that, the empirical mean of the clipped samples is a query with sensitivity $\Delta = \frac{2B}{n}$. For $\frac{\varepsilon^2}{2}$-zCDP, we need $\sigma \geq \frac{\Delta}{\varepsilon} = \frac{2B}{\varepsilon n}$ which is exactly the same as required in the proof of Ghazi et al. Thus the algorithm achieves the exact same sample complexity of $\tilde{O}\left(\frac{R+\sqrt{d}}{\varepsilon}\right)$ while satisfying $\frac{\varepsilon^2}{2}$-zCDP. ∎

**Corollary 22 (Unbounded Mean)** *There exists an $\frac{\varepsilon^2}{2}$-zCDP algorithm that performs $\alpha$-sampling for $N^d(\leq \infty, I)$ with sample complexity $\tilde{O}\left(\frac{\sqrt{d}}{\varepsilon}\right)$.*

**Proof** This follows from theorem 21, the pure DP reduction of lemma 11 with $\kappa = 1$ and the fact that any pure DP algorithm also satisfies zCDP. ∎

## 5. Lower Bounds for Multi-Sampling

### 5.1. A Lower Bound for Strong Multi-sampling

We will use a recent result by Kontorovich (2024):

**Theorem 23** *For any two sequences of probability distributions $\mathbf{D}_1, \ldots, \mathbf{D}_m$ and $\hat{\mathbf{D}}_1, \ldots, \hat{\mathbf{D}}_m$ on finite sets, if $\Delta$ is the vector of TV-distances $(||\mathbf{D}_1 - \hat{\mathbf{D}}_1||_{TV}, \ldots, ||\mathbf{D}_m - \hat{\mathbf{D}}_m||_{TV})$ then*

$$||\mathbf{D}_1 \otimes \cdots \otimes \mathbf{D}_m - \hat{\mathbf{D}}_1 \otimes \cdots \otimes \hat{\mathbf{D}}_m||_{TV} \geq c \cdot \min(1, ||\Delta||_2)$$

*where $c > 0.1798$ is a universal constant.*

We will use this to argue that any strong multi-sampler is in essence performing highly accurate single-sampling.

**Theorem 24** *Fix $\alpha < 0.1798$. Let $M$ be a strong $(m, \alpha)$-sampler for $\mathcal{D}$ with sample complexity $n$. There exists $M'$ an $\alpha/\sqrt{m}$-sampler for $\mathcal{D}$ with sample complexity $n$. Moreover, if $M$ ensures $(\varepsilon, \delta)$-DP then $M'$ also ensures $(\varepsilon, \delta)$-DP.*

**Proof** $M'$ is the algorithm that executes $M$ on its input and reports the first of the $m$ outputs of $M$. The DP guarantee is immediate from post-processing.

Suppose for contradiction that $M'$ is not an $\alpha/0.1798\sqrt{m}$-sampler. This means, for some $\mathbf{D} \in \mathcal{D}$ that generates the $n$ i.i.d. inputs, the distribution from which its output comes from is some $\hat{\mathbf{D}}$ where $||\hat{\mathbf{D}} - \mathbf{D}||_{TV} > \alpha/0.1798\sqrt{m}$. Via Theorem 23, we know $||\hat{\mathbf{D}}^{\otimes m} - \mathbf{D}^{\otimes m}||_{TV} \geq c \cdot \min(1, \sqrt{m}||\hat{\mathbf{D}} - \mathbf{D}||_{TV}) > 0.1798 \cdot \min(1, \sqrt{m}||\hat{\mathbf{D}} - \mathbf{D}||_{TV}) > \min(0.1798, \alpha) = \alpha$. But we also know $\hat{\mathbf{D}}^{\otimes m}$ is precisely the output distribution of $M$ because its output is i.i.d., so we have a contradiction with the accuracy of $M$. ∎

This result implies lower bounds on the sample complexity of strong multi-sampling. We instantiate it for $k$-ary distributions, via the lower bound on single-sampling by Raskhodnikova et al. (2021):

**Theorem 25** *For sufficiently small $\alpha > 0$, any strong $(m, \alpha)$-sampler for $k$-ary distributions must have sample complexity $n = \Omega\left(\sqrt{m} \cdot \frac{k}{\alpha\varepsilon}\right)$.*

Note that this separates strong multi-sampling from weak multi-sampling in the regime where $m = \Omega\left(\frac{1}{\varepsilon^2} \log^2 \frac{1}{\delta}\right)$ and $k = \Omega(\log \frac{1}{\delta})$ (compare with Theorem 13).

### 5.2. A Generic Recipe for Weak Multi-Sampling Lower Bounds

Thus far, we have assumed that DP algorithms ensure privacy for all inputs. But there exist algorithms $A$ which consume inputs for whom DP is not enforced. Specifically, there are $n$ inputs labeled "private" and $m$ inputs labeled "public" such that $A$ is insensitive to any change in any one of the "private" inputs. We refer to $m$ as the "public" sample complexity and $n$ as the "private" sample complexity.

We will use such *semi-private* [3] algorithms to derive lower bounds.

Given a task $T$ concerning distribution $\mathbf{D} \in \mathcal{D}$, suppose that the two statements below are true

1. If a DP algorithm performs $T$ without public samples, then it has private sample complexity $> \tau$.

2. There is a semi-private algorithm that performs $T$ by consuming $n^* < \tau$ private samples from $\mathbf{D}$ and $m^*$ public samples *from a different* $\hat{\mathbf{D}}$, where $||\mathbf{D} - \hat{\mathbf{D}}||_{TV} \leq \alpha$.

If we had a weak $(m = m^*, \alpha)$-sampler for $\mathcal{D}$ with sample complexity $n < n^*$, we can treat the generated variables as the public samples required by the algorithm in (2) to do $T$. But overall we have only consumed $< \tau$ private samples from $\mathbf{D}$, which means we have arrived at a contradiction.

Via Lemma 8, our recipe also applies to single-sampling: if we had a single-sampler with sample complexity $n < n^*/m$, there is a weak $(m, \alpha)$-sampler with sample complexity $< n^*$.

### 5.3. Lower Bound for Weak Multi-Sampling of Bounded-Covariance Gaussians

We define the Gaussian learning task.

**Definition 26 (Learning Bounded Gaussians)** *An algorithm learns $N^d(\leq R, \preceq \kappa)$ up to error $\alpha$ if it can consume independent samples from $\mathbf{D} \in N^d(\leq R, \preceq \kappa)$ and produce some $\tilde{\mathbf{D}} := N(\tilde{\mu}, \tilde{\Sigma})$ where $||\mathbf{D} - \tilde{\mathbf{D}}||_{TV} \leq \alpha$ with 90% probability, where the randomness is over the algorithm and the samples.*

Now, we instantiate point 1 of our recipe with the following lower bound sketched by Kamath et al.:

**Theorem 27 (Lower bound for zCDP learning of bounded Gaussians Kamath et al. (2019))** *If an algorithm learns $N^d(\leq R, \preceq \kappa)$ up to error $\alpha$ and enforces $\frac{\varepsilon^2}{2}$-zCDP on all inputs, then it must have a sample complexity $\Omega(\frac{\sqrt{d}}{\varepsilon}\sqrt{\log R} + \frac{1}{\varepsilon}\sqrt{\log \kappa})$*

We instantiate point 2 of our recipe with the following result by Bie et al.:

---

3. The term comes from the work by Beimel et al. (2013).

**Theorem 28 (Upper bound for semi-private zCDP learning of Gaussians Bie et al. (2022))** *There is a computationally efficient semi-private algorithm that consumes $n^* = \tilde{O}\left(\frac{d^2}{\alpha^2} + \frac{d^2}{\alpha\varepsilon}\right)$ private samples from a $d$-dimensional Gaussian $\mathbf{D}$ and $m^* = d + 1$ public samples from a Gaussian $\hat{\mathbf{D}}$ with $||\mathbf{D}, \hat{\mathbf{D}}||_{TV} < \alpha$ to perform Gaussian learning while ensuring $\frac{\varepsilon^2}{2}$-zCDP of the private samples.*

Finally, we derive our lower bound.

**Theorem 29 (Lower bound on zCDP weak multi-sampling)** *If an $\frac{\varepsilon^2}{2}$-zCDP algorithm is a weak $(d + 1, \alpha)$-sampler for $N^d(\leq R, \preceq \kappa)$, then it has sample complexity*

$$\tilde{\Omega}\left(\left(\frac{d^2}{\alpha^2} + \frac{d^2}{\alpha\varepsilon}\right) + \frac{\sqrt{d}}{\varepsilon}\sqrt{\log R} + \frac{1}{\varepsilon}\sqrt{\log\kappa}\right)$$

**Proof** Let $c_\ell, c_u$ be leading constants in Theorems 27 and 28 respectively. Let $p(d, \alpha, \varepsilon)$ be the polylogarithmic term in Theorem 28. We claim that, for all sufficiently large $d, R, \kappa, 1/\alpha, 1/\varepsilon$, the function

$$s(d, R, \kappa, \alpha, \varepsilon) := c_u \cdot \left(\frac{d(d+1)}{\alpha^2} + \frac{d(d+1)}{\alpha\varepsilon}\right) \cdot p(d, \alpha, \varepsilon) + \frac{c_\ell}{2} \cdot \left(\frac{\sqrt{d}}{\varepsilon}\sqrt{\log R} + \frac{1}{\varepsilon}\sqrt{\log\kappa}\right)$$

is a lower bound on the sample complexity. If it were not the case, we can use $s$ private samples from $\mathbf{D}$ to generate $d + 1$ samples from a Gaussian that is $\alpha$-close to $\mathbf{D}$ (under zCDP). Then we can use Bie et al.'s algorithm to learn $\mathbf{D}$ by feeding it $n^* < s$ of the private samples and all $d + 1$ generated samples.

Now, for sufficiently large $R$ and $\kappa$,

$$s < c_\ell \cdot \left(\frac{\sqrt{d}}{\varepsilon}\sqrt{\log R} + \frac{1}{\varepsilon}\sqrt{\log\kappa}\right).$$

But this contradicts the lower bound on zCDP learning without public data. ∎

## Acknowledgments

We thank Sofya Raskhodnikova, Adam Smith and Satchit Sivakumar for introducing us to this problem and for providing helpful comments. We also thank Rasmus Pagh for letting us know about the Generalized Gaussian Mechanism.

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

## Appendix A. Reduction from Unbounded Mean to Bounded Mean for Single Sampling of Gaussians

**Lemma 11 (Reduction for Unbounded Mean)** *For any $d \geq 1, R > 0, \varepsilon > 0, \delta \in [0,1)$, and $\alpha, \delta' \in (0,1)$. If there exists an $(\varepsilon, \delta \geq 0)$-DP algorithm for $\alpha$-sampling of $N^d(\leq R, \preceq \kappa)$ with sample complexity $n(d, R, \alpha, \varepsilon, \delta)$, then*

- *(Pure DP reduction). There exists an $(2\varepsilon, \delta)$-DP algorithm for $2\alpha$-sampling of $N^d(\leq \infty, \preceq \kappa)$ with sample complexity $n(d, O(\kappa\sqrt{d}), \alpha, \varepsilon, \delta) + \tilde{O}(d \log(d)/\varepsilon)$.*

- *(Approximate DP reduction). There exists an $(2\varepsilon, \delta + \delta')$-DP algorithm for $2\alpha$-sampling of $N^d(\leq \infty, \preceq \kappa)$ with sample complexity $n(d, O(\kappa\sqrt{d}), \alpha, \varepsilon, \delta) + \tilde{O}(\sqrt{d} \log(d)/\varepsilon)$.*

*The same holds for both variants of $(m, \alpha)$-sampling.*

**Proof** [Proof Sketch of Lemma 11]

**Accuracy Analysis.** The *DensestBall* algorithm by Ghazi et al. (2020, Theorem 6) is the main ingredient of this reduction. If there is a ball of radius $r$ that contains a majority of the dataset, then *DensestBall* given $r$ as input, outputs a ball of radius $O(r)$ that also contains a majority of the dataset. Since $\Sigma \preceq \kappa \cdot I$, we can use a concentration inequality for Gaussians to set $r = O(\kappa\sqrt{d})$. Let $c$ be the center of the ball that *DensestBall* outputs. This algorithm has a guarantee that $||\mu - c|| \leq r$ holds with high probability. Shifting each input sample points, $X$ to $X - c$, reduces to the case where $||\mu|| \leq r$ i.e., the mean is bounded.

**Privacy Analysis.** The *DensestBall* algorithm has both pure DP and approximate DP variants. Let us assume we run the pure DP (approximate DP) variant with privacy budget $\varepsilon$ (or $(\varepsilon, \delta')$ respectively). We are given that $(\varepsilon, \delta)$-DP algorithm for the bounded mean case exists. Using basic composition gives us that the algorithm for the unbounded mean case is $(2\varepsilon, \delta)$-DP (or $2\varepsilon, \delta + \delta')$-DP respectively.

The pure DP variant of *DensestBall* has a sample complexity of $\tilde{O}(d \log(d)/\varepsilon)$ whereas the approximate DP variant has a sample complexity of $\tilde{O}(\sqrt{d} \log(d)/\varepsilon)$ which is reflected in the final sample complexity. ∎

## Appendix B. DP Weak Multi-Sampling for Finite Distributions, with Hints

Previously, we assumed that all distributions in consideration $\mathbf{D} \in \mathcal{D}$ have support $S$ where $S \subseteq [k]$. Here, we consider cases where other information about $S$ is known. We effectively reduce to the known support case by using established algorithms to identify a superset of the support.

The central idea is that as long as we can find an interval which contains more than $1 - \alpha$ fraction of the probability mass of the distribution $\mathbf{D}$, we can run ShuRR on that interval.

### B.1. Contiguous and Bounded Support

Some contiguous interval of width $k$ contains $S$. In this case, we use the classic stability-based histogram algorithm of Bun et al. (2016)—which we call *noise-and-threshold*—to find the mode $v$ of $\mathbf{D}$ and then run ShuRR assuming $[v - k, v + k]$ is the support. The number of samples to find $v$ is $O(\frac{k}{\varepsilon} \log \frac{1}{\delta})$.

---

**Algorithm 5:** Multi-Sampler for Contiguous and Bounded Support

---

**Parameters**: $\varepsilon, \delta > 0$
**Input:** $n$ samples $X_1, \ldots X_n$ from a distribution whose support lies in some contiguous interval
      of width $k$
**Output:** $m$ samples $\hat{X}_1, \ldots, \hat{X}_m$
Run the *noise-and-threshold* algorithm using the first $O(\frac{k}{\varepsilon} \log \frac{1}{\delta})$ input samples.
Let $T \subset \mathbb{Z}$ be the set of integers that the above identified as having nonzero frequency
**if** $T = \emptyset$ **then**
  |   **return** $\perp$
**end**
Let $v$ be the integer in $T$ with the largest estimated frequency
**return** *ShuRR*$(X)$, *with support* $[v - k, v + k]$

---

## B.2. Bounded Support

We assume only $|S| \leq k$.

Option 1 (non-adaptive): Run the *noise-and-threshold* algorithm to identify a subset $E$ of the support, such that the items that are missing each have mass at most $\alpha/k$ so that the distribution conditioned on $E$ is within $\alpha$ of the original. The number of samples to do this is $O(\frac{k}{\alpha\varepsilon} \log(k/\delta))$

---

**Algorithm 6:** Multi-Sampler for Bounded Support Size

---

**Parameters**: $\varepsilon, \delta > 0, \alpha$
**Input:** $n$ samples $X_1, \ldots X_n$ from a distribution whose support size $|S| \leq k$.
**Output:** $m$ samples $\hat{X}_1, \ldots, \hat{X}_m$
Run the *noise-and-threshold* algorithm using the first $O(\frac{k}{\alpha\varepsilon} \log \frac{k}{\delta})$ input samples.
Let $E \subset \mathbb{Z}$ be the set of integers that the above identified as having nonzero frequency
**if** $E = \emptyset$ **then**
  |   **return** $\perp$
**end**
**return** *ShuRR*$(X)$, *with support* $E$

---

Option 2 (adaptive): Find an element $v$ from the support of $\mathbf{D}$ using the *noise-and-threshold* algorithm, then run the *above-threshold* algorithm to find $h$ such that $[v - h, v + h]$ contains $1 - O(\alpha)$ fraction of samples, then run ShuRR assuming $[v - h, v + h]$ is the support. The number of samples to find $v$ is $O(\frac{k}{\varepsilon} \log \frac{1}{\delta})$. The number of samples to find $h$ is $O(\frac{1}{\varepsilon} \log \frac{w}{\alpha})$, where the value of $w$ is determined within the algorithm. The number of samples given to ShuRR is $O(m + \frac{h}{\alpha\varepsilon} \log \frac{1}{\delta})$

## B.3. Known width of confidence interval

Some contiguous interval of width $w$ contains both the mode $v$ and $1 - O(\alpha)$ of the mass of $\mathbf{D}$.

For example, consider shifted binomials, $w \approx \sqrt{k \log(1/\alpha)}$ ). In this case, we use $[v - w, v + w]$ as the support. The number of samples to find $v$ is $O(\frac{w}{\varepsilon} \log \frac{1}{\delta})$ and the number of samples consumed by ShuRR is $O(m + \frac{w}{\alpha\varepsilon} \log \frac{1}{\delta})$. If we also knew $|S| \leq k$ then $v$ can be found with $O(\frac{\min w, k}{\varepsilon} \log \frac{1}{\delta})$.

---

**Algorithm 7:** ShuRR$^+$

---

Run the noise-and-threshold algorithm on $X$.

Let $T \subset \mathbb{Z}$ be the set of integers that the above identified as having nonzero frequency in $X$

**if** $T = \emptyset$ **then**

|     **return** $\perp$

**end**

Let $v$ be the integer in $T$ with the largest estimated frequency

**for** $w \in \{16, 32, 64, 128, \dots\}$ **do**

    Let $Y$ be $\frac{8}{\varepsilon\alpha} \ln \frac{2w}{\alpha}$ new i.i.d. samples from $\mathbf{D}$

    Run *above-threshold* (Dwork and Roth, 2014) on $Y$ with threshold $1 - \alpha$ & queries

    $f_0, f_1, \dots, f_w$, where $f_\Delta$ is the fraction of $Y$ that lie in $[v - \Delta, v + \Delta]$

    Break out of for-loop if *above-threshold* responded $\top$ at some iteration $h$

**end**

Let $Z$ be $m + O(\frac{h}{\varepsilon^2} \log \frac{1}{\delta})$ new i.i.d. samples from $\mathbf{D}$

**return** *ShuRR$(Z)$, with support* $[v - h, v + h]$

---

## B.4. Sub-Gaussian distribution

In this case, the distribution $\mathbf{D}$ is a discrete sub-Gaussian distribution over $\mathbb{Z}$ with known variance proxy $\sigma$.

Due to the sub-Gaussian property of $\mathbf{D}$, most of the probability mass is concentrated in an interval $I$ of size $2\sigma$ centered at the true mean $\mu$ of $\mathbf{D}$. In our algorithm, we divide $\mathbb{Z}$ into bins of size $O(\sigma)$. The interval $I$ mentioned above will intersect at most 2 bins. Running the *noise-and-threshold* using these bins will help us identify these two bins giving us a rough estimate of where $\mu$ lies. Now, we can extend this rough estimate of where $\mu$ lies on both sides by $O(\sigma \log 1/\alpha)$ to create a new interval which contains at least $1 - \alpha$ fraction of the probability mass of $\mathbf{D}$. We then run ShuRR using this new interval.

## B.5. Techniques

### Finding at least one element from the support of a distribution

**Lemma 30** *Given $X$ a set of $n = O(\frac{k}{\varepsilon} \log \frac{1}{\delta})$ i.i.d samples from a distribution $\mathbf{D}$ with support size $\leq k$, the probability that the noise-and-threshold algorithm finds at least one element from the support with non-zero frequency is at least $1 - 2\delta$.*

**Proof** No element from the support will be reported if *noise-and-threshold* erases all elements of the support that were present among the the $n$ input samples. An element will be dropped if its noised count falls beneath a threshold $t = O(\frac{1}{\varepsilon} \log \frac{1}{\delta})$. So it suffices to argue that there is some element of the support that has noisy count greater than $t$, except with probability $\leq 2\delta$.

Observe that the mode of $\mathbf{D}$ has probability mass $\geq 1/k$. By a multiplicative Chernoff bound, our choice of $n$ ensures the mode will have count exceeding $2t$ in $X$, except with probability $\leq \delta$. We also know that adding Laplace noise to $2t$ will result in a value less than $t$, except with probability $\leq \delta$. A union bound completes the proof. ∎

### Finding all the heavy elements from the support of a distribution

Given a distribution $\mathbf{D}$, we define *heavy* elements from the support of $\mathbf{D}$ as elements which have probability mass at least $\alpha/k$.

**Lemma 31** *Given $X$ a set of $n = O(\frac{k}{\alpha\varepsilon} \log \frac{1}{\delta})$ i.i.d samples from a distribution $\mathbf{D}$ with support size $\leq k$, the probability that the noise-and-threshold algorithm run with privacy parameters $\varepsilon, \delta/k$ finds every heavy element of $\mathbf{D}$ with non-zero frequency is at least $1 - 2\delta$.*

**Proof** Consider any arbitrary heavy element $e$ in the support of $\mathbf{D}$. For *noise-and-threshold* to report $e$, the noisy count of $e$ must be non-zero. We know an element will be dropped if it's count is less than the threshold of $t = O(\frac{1}{\varepsilon} \log \frac{k}{\delta})$

Notice that, since $e$ is a heavy element, it's probability mass is at least $\alpha/k$. Using the same logic as in the previous lemma, we can use a multiplicative Chernoff bound to show that our choice of $n$ ensures that $e$ will have count which exceeds $2t$ in $X$. except with probability $\delta/k$. We also know that adding Laplace noise to $2t$ will result in a value less than $t$, except with probability $\leq \delta/k$. A union bound over these two events, tells us that *noise-and-threshold* will find $e$, except with probability $2\delta/k$.

There can be at most $k$ heavy elements in the support, as the support size $\leq k$. Union bounding over at most $k$ heavy elements completes the proof. ∎

## Appendix C. Weak Multi-Sampling for k-ary distributions using **ShuRR**

**Theorem 13** *There exists a choice of $f : \mathbb{R}^+ \to \mathbb{R}^+$ such that, for any $0 < \varepsilon, \delta, \alpha < 1$ and $m \in \mathbb{N}$, ShuRR is $(\varepsilon, \delta)$-DP and performs weak $(m, \alpha)$-sampling with sample complexity $n = O\left(m + \frac{k}{\alpha\varepsilon^2} \log \frac{1}{\delta}\right)$. There is another $f$ which ensures $\varepsilon$-DP for $\varepsilon > 1$; the sample complexity becomes $O\left(m + \frac{k}{\alpha\varepsilon} \log \frac{1}{\delta}\right)$.*

For shuffled randomized response, Feldman et al. (2021) express the privacy parameter of the shuffled output as a function of $\varepsilon_0$ and target $\delta$:

**Lemma 32 (Amplification-by-Shuffling of Rand. Response Feldman et al. (2021))** *ShuRR guarantees $(\varepsilon_1, \delta)$-DP, where*

$$\varepsilon_1 = \log\left(1 + 8(\exp(\varepsilon_0) + 1)\left(\sqrt{\frac{k+1}{k} \cdot \frac{\log 4/\delta}{n} \cdot \frac{1}{\exp(\varepsilon_0) + k - 1}} + \frac{k+1}{kn}\right)\right).$$

Our algorithm sets $\varepsilon_0$ such that this $\varepsilon_1$ bound is itself bounded by the desired $\varepsilon$.

**Proof** [Proof of Thm. 13] To bound the privacy parameter, we first apply the bound $\log(1 + x) \leq x$:

$$\varepsilon_1 \leq 8(\exp(\varepsilon_0) + 1)\left(\sqrt{\frac{k+1}{k} \cdot \frac{\log 4/\delta}{n} \cdot \frac{1}{\exp(\varepsilon_0) + k - 1}} + \frac{k+1}{kn}\right)$$

$$\leq 8(\exp(\varepsilon_0) + 1)\left(\sqrt{\frac{3}{2} \cdot \frac{\log 4/\delta}{n} \cdot \frac{1}{\exp(\varepsilon_0) + 1}} + \frac{3}{2n}\right) \qquad (k \geq 2)$$

$$= 8\left(\sqrt{\frac{3}{2} \cdot \frac{\log 4/\delta}{n} \cdot (\exp(\varepsilon_0) + 1)} + \frac{3}{2n}(\exp(\varepsilon_0) + 1)\right)$$

$$= 8 \cdot \left(\sqrt{\frac{3}{2} \cdot \frac{\log 4/\delta}{n} \cdot \frac{f^2(\varepsilon) \cdot n}{\ln(4/\delta)}} + \frac{3}{2n} \cdot \frac{f^2(\varepsilon) \cdot n}{\ln(4/\delta)}\right) \qquad \text{(Choice of } \varepsilon_0)$$

$$= 8\sqrt{3/2} \cdot f(\varepsilon) + \frac{12}{\ln(4/\delta)} \cdot f^2(\varepsilon)$$

When $f(\varepsilon) = \varepsilon/(16\sqrt{3/2})$ and $\varepsilon < 1$, the above is bounded by $\varepsilon/2 + \varepsilon^2/32 < \varepsilon$. When $f(\varepsilon) = \sqrt{\varepsilon}/(16\sqrt{3/2})$ and $\varepsilon > 1$, it is bounded by $\sqrt{\varepsilon}/2 + \varepsilon/32 < \varepsilon$.

It remains to prove that that Algorithm 2 performs weak multi-sampling. Identical to the prior theorem, the TV distance between $\mathbf{D}$ and the outcome of randomized response on $X \sim \mathbf{D}$ is the mixture weight $\frac{k-1}{k-1+\exp(\varepsilon_0)}$. By substituting the new $\varepsilon_0$ value, we have

$$\frac{k-1}{k-1+\exp(\varepsilon_0)} = \frac{k-1}{k-1+\frac{f^2(\varepsilon)\cdot n}{\ln(4/\delta)} - 1} = \frac{k-1}{k + \frac{f^2(\varepsilon)\cdot n}{\ln(4/\delta)} - 2} \qquad (4)$$

We will bound the above by $\alpha$ in the regime where $n \geq \frac{k\ln(4/\delta)}{\alpha f^2(\varepsilon)}$.

$$(4) \leq \frac{k-1}{k + \frac{k}{\alpha} - 2} \leq \frac{k-1}{k/\alpha} < \alpha$$

The first inequality comes from substituting the bound on $n$. The second comes from the fact that $k \geq 2$.

Our final sample complexity bound is therefore $\max\left(m, \frac{k\ln(4/\delta)}{\alpha f^2(\varepsilon)}\right)$. The big-Oh bound in the theorem statement simply collapses the maximum into a summation. Also note that $1/f^2(\varepsilon)$ is $O(1/\varepsilon^2)$ for $\varepsilon < 1$ and $O(1/\varepsilon)$ otherwise. ∎

## Appendix D. Gaussians with Bounded Covariance

In this section, we show the existence of single sampling algorithms for Gaussians with bounded covariance i.e. $N^d(\leq \infty, \leq \kappa)$ under zCDP . We show that an approximate DP algorithm proposed by Ghazi et al. (2023, Algorithm 3) satisfies zCDP. Weak and strong multi-sampling algorithms can be derived by way of Lemmas 8 and 9.

**Theorem 33 (Bounded Mean)** *For any finite $\kappa > 0$, there exists an $\frac{\varepsilon^2}{2}$-zCDP algorithm that performs $\alpha$-sampling for $N^d(\leq R, \leq \kappa)$ with sample complexity $\tilde{O}\left(\frac{R^2\sqrt{d}+d^{3/2}}{\alpha\varepsilon^2}\right)$.*

---

**Algorithm 8:** Bounded Covariance Gaussian Single-Sampler (Ghazi et al., 2023, Algorithm 3)

---

**Input:** $X_1, \ldots, X_{n_1}, X_{n_1+1}, \ldots, X_{n_1+2n_2} \in \mathbb{R}^d$
**Output:** $y \in \mathbb{R}^d$
**Parameters:** $B, \sigma > 0, n_1, n_2 \in \mathbb{N}$
$\{\hat{X}_i \leftarrow X_i \cdot \min\left(\frac{B}{\|X_i\|_2}, 1\right)\}_{i \in [n_1+2n_2]}$
$Z \sim N(0, \sigma^2 \cdot I)$
$y \leftarrow Z + \frac{1}{n_1}\left(\sum_{i \in [n_1]} X_i\right) + \sqrt{\frac{1-1/n_1}{2n_2}}\left(\sum_{i \in [n_2]}(X_{n_1+2i-1} - X_{n_1+2i})\right)$
**return** $y$

---

Ghazi et al. (2023) provide an optimal approximate DP algorithm (their Algorithm 2) to perform single-sampling for $N^d(\leq R, \leq \kappa)$ with sample complexity $\tilde{\Theta}\left(\frac{d}{\varepsilon}\right)$. However, this algorithm uses the "propose-test-release" paradigm of Dwork and Lei (2009) and can't be $\varepsilon^2/2$-zCDP. They also provide a separate approximate DP algorithm (their Algorithm 3) for the same problem which uses the Gaussian Mechanism with a worse sample complexity of $\tilde{O}\left(\frac{d^{3/2}}{\alpha\varepsilon^2}\right)$ (Ghazi et al., 2023, Theorem C.1). We show that it satisfies zCDP.

**Proof** We will use the same set of parameters that Ghazi et al. used. In particular, let $n_1 = n_2$, $B = \tilde{O}\left(R + \sqrt{d}\right)$ and $\sigma^2 = \frac{\alpha}{4\sqrt{d}}$. Note that the accuracy analysis proceeds exactly the same way as presented in Ghazi et al. For the privacy analysis, the final expression being returned in the algorithm, before adding the noise $Z \sim N(0, \sigma^2 I)$, has a global sensitivity $\Delta = \frac{B}{\sqrt{n}}$. For $\frac{\varepsilon^2}{2}$-zCDP, we need

$$\sigma^2 \geq \frac{\Delta^2}{\varepsilon^2} \implies \frac{\alpha}{4\sqrt{d}} \geq \frac{B^2}{n\varepsilon^2} \implies n \geq \frac{4\sqrt{d}B^2}{\alpha\varepsilon^2} \implies n = \tilde{O}\left(\frac{R^2\sqrt{d} + d^{3/2}}{\alpha\varepsilon^2}\right).$$

∎

**Corollary 34 (Unbounded Mean)** *For any finite $\kappa > 0$, there exists an $\frac{\varepsilon^2}{2}$-zCDP that performs $\alpha$-sampling for $N^d(\leq \infty, \leq \kappa)$ with sample complexity $\tilde{O}\left(\frac{d^{3/2}}{\alpha\varepsilon^2}\right)$.*

**Proof** This follows from theorem 33, the pure DP reduction of lemma 11, and the fact that any pure DP algorithm also satisfies zCDP. ∎

## Appendix E. The Euclidean-Laplace distribution

### E.1. Validity of density function

**Lemma 35** *For any scale parameter $b > 0$ and dimensionality $d \in \mathbb{N}$, the Euclidean-Laplace's density function integrates to 1.*

**Proof** Recall that the density function is

$$F_{\mathbf{ELap}(b)}(\eta) = \frac{\Gamma(d/2)}{2\pi^{d/2}b^d\Gamma(d)} \cdot \exp\left(-\frac{\|\eta\|_2}{b}\right)$$

We will perform the integral along polar coordinates. That is to say, we integrate over the surface of the $(d-1)$-sphere of radius $r$ and then integrate over all $r > 0$. For neatness, we define

$$f(r) := \frac{\Gamma(d/2)}{2\pi^{d/2}b^d\Gamma(d)} \cdot \exp\left(-\frac{r}{b}\right).$$

So the integral is

$$\int_{r>0} f(r) \cdot \underbrace{r^{d-1} \cdot \frac{2\pi^{d/2}}{\Gamma(d/2)}}_{\text{surface area}} \, \mathrm{d}r$$

$$= \frac{1}{b^d\Gamma(d)} \cdot \int_{r>0} \exp(-r/b) \cdot r^{d-1} \mathrm{d}r \tag{5}$$

Let $t := r/b$ so that $\mathrm{d}r = b\,\mathrm{d}t$. By substitution,

$$(5) = \frac{1}{b^d\Gamma(d)} \int_{t=0}^{\infty} \exp(-t) \cdot (b^{d-1}t^{d-1}) \cdot b\,\mathrm{d}t$$

$$= \frac{1}{\Gamma(d)} \int_{t=0}^{\infty} t^{d-1}e^{-t}\,\mathrm{d}t$$

$$= 1$$

The last step comes from the definition of the Gamma function. ∎

### E.2. Sampling algorithm

**Lemma 1** *Suppose there exists a Gaussian oracle that, when given $\sigma > 0$, produces a sample from $N(0, \sigma^2)$. Also suppose there exists a Gamma oracle that, when given shape parameter $k > 0$ and scale parameter $\theta > 0$, produces a sample from $\Gamma(k, \theta)$. Then there exists an algorithm that, when given $b > 0$, produces a sample from $d$-dimensional $\mathbf{ELap}(b)$ by making 1 call to the Gamma oracle and $d$ calls to the Gaussian oracle.*

**Proof** The algorithm is simple:

1. Sample the radius $r$ from the Gamma distribution with parameters $k = d$ and $\theta = b$.

2. Sample $\{v_i\}_{i\in[d]}$ independently from $N(0, 1)$.

3. Create $\hat{v} := \frac{v}{\|v\|_2}$, a uniformly random point on the unit sphere in $d$ dimensions.

4. Output $r\hat{v}$

Now we need to show that $r\hat{v}$ is distributed as $\mathbf{ELap}(b)$. As implied by (5) in the preceding proof, the distribution of the $\ell_2$ norm of $\eta \sim \mathbf{ELap}(b)$ has density

$$\frac{1}{b^d\Gamma(d)} \cdot \exp(-r/b) \cdot r^{d-1}$$

at value $r > 0$. But also note that the Gamma distribution with parameters $k$ and $\theta$ has density

$$\frac{x^{k-1}e^{-x/\theta}}{\Gamma(k)\theta^k}$$

at value $x > 0$. The two are isomorphic.

The proof is complete by observing that, conditioned on any norm $r$, the distribution of an $\textbf{ELap}(b)$ variable is spherically symmetric. ∎

### E.3. Tail bound

We focus on bounding the tail of the norm of a Euclidean-laplace random variable:

**Lemma 17** *For any $\alpha \in (0, 1)$ and $\eta \sim \textbf{ELap}(b)$, $\mathbb{P}\left[\|\eta\|_2 > db \ln \frac{d}{\alpha}\right] \leq \alpha$.*

First, we prove that norm of a E-L r.v. follows a Gamma distribution. Then recall how to express a Gamma random variable as a sum of exponential random variables. Finally, we employ a tail bound on that sum.

**Lemma 36** *Let $\eta \in \mathbb{R}^d$ such that $\eta \sim \textbf{ELap}(b)$, then $\|\eta\|_2 \sim Gamma(d, 1/b)$*

**Proof** $F_{\textbf{ELap}(b)}(\eta)$ depends on the norm of $\eta$ i.e. $\|\eta\|_2$ and is independent of the direction $\hat{\eta}$. Thus the probability density of the norm $\|\eta\|_2$ at distance $r$ from the origin, denoted by $F(r)$, can be found by integrating $F_{\textbf{ELap}(b)}(\eta)$ over all $\eta$ that lie on the surface of $d - 1$ dimensional sphere of radius $r$ denoted by $S_{d-1}(r)$. Since the pdf $F_{\textbf{ELap}(b)}$ is constant over the surface of the sphere $S_{d-1}(r)$, $F(r)$ is equal to the probability density of $\eta$ where $\|\eta\|_2 = r$ times the surface area of $S_{d-1}(r)$.

$$F(r) = \underbrace{\frac{\Gamma(d/2)}{2\pi^{d/2}b^d\Gamma(d)} \cdot \exp\left(-\frac{r}{b}\right)}_{\text{probability density } F_{\textbf{ELap}(b)} \text{ at distance } r} \times \underbrace{\frac{2\pi^{d/2}r^{d-1}}{\Gamma(d/2)}}_{\text{surface area of } S_{d-1}(r)} = \frac{r^{d-1}}{\Gamma(d)b^d}\exp\left(-\frac{r}{b}\right)$$

Notice that the pdf above is the pdf of the Gamma distribution with shape paramter $d$ and rate parameter $1/b$. ∎

**Fact 37** *If $X \sim Gamma(k, \lambda)$ where $k$ is the shape parameter and $\lambda$ is the rate parameter then $X$ can be expressed as the sum of $k$ independent and identically distributed random variables $X_1, \ldots, X_k$ where each $X_i$ follows an exponential distribution with paramter $\lambda$ i.e. $X_i \sim Exp(\lambda)$ for all $i \in [k]$.*

**Lemma 38** *Let $X \sim Gamma(k, \lambda)$ then,*

$$\Pr(X > t) \leq k \cdot \exp\left(-\frac{\lambda t}{k}\right)$$

**Proof** We know from fact 37 that $X$ can be decomposed into independent and identically distributed random variables $X_1, \ldots X_k$ such that each $X_i \sim Exp(\lambda)$. For $X$ to be greater than the threshold $t$, at least one of the $k$ $X_i$'s has to be greater than $t/k$ by a simple averaging argument. Thus,

$$\Pr\left(X > t\right) \leq \Pr\left(\text{At least one } X_i > t/k\right)$$

The probability that $X_i > t/k$ can be calculated from its CDF expression.

$$\Pr\left(X_i > t/k\right) = 1 - \Pr\left(X_i \leq t/k\right) = 1 - CDF_{Exp(\lambda)}(t/k) = 1 - \left(1 - \exp\left(-\frac{\lambda t}{k}\right)\right) = \exp\left(-\frac{\lambda t}{k}\right)$$

Union bounding over all the $k$ random variables, we get

$$\Pr\left(\text{At least one } X_i > t/k\right) \leq k \cdot \Pr\left(X_i > t/k\right) = k \cdot \exp\left(-\frac{\lambda t}{k}\right)$$

Thus, we get

$$\Pr\left(X > t\right) \leq k \cdot \exp\left(-\frac{\lambda t}{k}\right)$$

∎

We are now ready to complete our proof.

**Proof** [Proof of Lemma 17] Lemma 36 tells us that $\|\eta\|_2 \sim Gamma(d, 1/b)$. Using Lemma 38, we get that

$$\Pr\left(\|\eta\|_2 > t\right) \leq d \cdot \exp\left(-\frac{t}{db}\right)$$

Substituting $t = db \log\left(\frac{d}{\alpha}\right)$ completes the proof. ∎

