# OpenReview forum: "Differentially Private Multi-Sampling from Distributions"
_algorithmiclearningtheory.org/ALT/2025/Conference — ALT 2025_

### Official Review · Reviewer_Z8Uc · 2024-11-08
**Good paper on new notions of private sampling**

**Rating:** 6
**Confidence:** 3

**Review:**

This paper proposes and studies multi-sampling, which generalizes the notion of single sampling proposed in the previous papers. Essentially, given i.i.d samples from distribution D, multi-sampling asks the algorithm to output m samples, with distribution close to $D^{\otimes m}$ .
Despite the question has been studied when m=1, running the algorithm multiple times does not yield the right sample complexity even for very simple distributions, as shown in the paper.

In the paper, they study two basic distributions, the discrete k-ary distribution and the Gaussian distribution.
For discrete k-ary distribution, under approximate differential privacy, they give improved sample complexity linear in $m/\epsilon$, which improves the trivial baseline linear in $m^2/\epsilon$, where $\epsilon$ is the privacy parameter.
For Gaussian distribution, they give pure differential privacy algorithms, while previously only approximate DP algorithms are known.

**Merits** The problem studied is natural and exciting, especially due to the improvement over the baseline by running single sample algorithms.

**Drawbacks**
1. There should be a statement of (informal) main theorem at the beginning of the paper, for the most important result
2. The lower bounds and upper bounds for the sample complexity does not match for discrete distributions.
3. The Euclidean-Laplace mechanism for the gaussian distribution(equation (3)) does not seem to run in polynomial time.

**Paper Award:**

No

---

### Official Review · Reviewer_gp62 · 2024-11-09
**Interesting extensions of prior work**

**Rating:** 7
**Confidence:** 4

**Review:**

Raskhodnikova et al. (2021) introduced the problem of privately generating a sample from an unknown target distribution given i.i.d. samples from it. The goal is to design an algorithm A with the following properties:

1. A takes as input a dataset S of n i.i.d. samples from an unknown distribution D over a domain X, where D belongs to a known family C of distributions.
2. A outputs a point x from X such that the distribution induced by A(S) is within a total variation distance of at most alpha from D.
3. A satisfies differential privacy with respect to S.

This problem is trivial without the privacy constraint, as the algorithm could simply output the first point in S. However, Raskhodnikova et al. (2021) demonstrated that with differential privacy, this can be as challenging as privately learning the distribution D itself. Ghazi et al. (2023) extended this work to additional families of target distributions (including multi-dimensional Gaussian distributions) and presented improved results for product distributions.

This paper explores an extension where the task is to output m samples rather than just a single sample (referred to as "multi-sampling"). The authors introduce two possible notions of utility: a "weak" variant and a "strong" variant. Both require that the output random variables are mutually independent and identically distributed. The weak variant requires that the marginal distribution of the output is close to the input distribution, while the strong variant requires that the product distributions are close.

The authors present transformations from the single-sample case to the weak-m-sample case, and from the weak to the strong-m-sample case. This allows them to adapt algorithms from prior work to the multi-sampling setting. Additionally, they provide improved constructions for distributions over [k] and Gaussian distributions (under different assumptions about the information available to the private mechanism). Finally, they leverage a recent result by Kontorovich (2024) to derive a negative result for strong multi-sampling.

The extension to "multi-sampling" is a natural and worthwhile direction. The algorithms and proofs presented are elegant.  One potential limitation is that the results, in hindsight, may appear somewhat straightforward.  Overall, this is a solid paper and I recommend it for acceptance to ALT.

**Paper Award:**

No

---

### Official Review · Reviewer_dCNx · 2024-11-09
**This paper considers the the sampling process with differential privacy guarantee. In the paper, the authors propose several algorithm focusing on the cases of single sampling, weak sampling and strong sampling with privacy gurantee and accuracy in total variation. The authors also provided lower bounds for the multi-sampling problems.**

**Rating:** 7
**Confidence:** 3

**Review:**

This paper considers the the sampling process with differential privacy guarantee. In the paper, the authors propose several algorithm focusing on the cases of single sampling, weak sampling and strong sampling with privacy gurantee and accuracy in total variation. The authors also provided lower bounds for the multi-sampling problems.

**Strength:**

1. The paper is easy to follow. The authors make a clear list of the various definitions in the paper.

2. The accuracy of the output sample is in the strong sense of the total variation distance.

3. The authors discuss in details of different regimes with different information on the input distribution.

**Weakness and Suggestions:**

1. The model is only based on the k-ary distributions and Gaussian distributions. I wonder if similar method is also applicable to other distributions. Is the TV distance is the bottleneck?

2. There are lots of materials on private histogram algorithm, where the algorithm takes iid data as input and output a distribution with approximate histogram distribution. Take arxiv.org/abs/0811.2501 for example, which measures accuracy in L2 and Kolmogrov distance. There are also private synthetic data algorithms which takes an input dataset and output a new dataset with similar distribution, e.g. arxiv.org/abs/2302.05552. It seems that private sampling (especially weak multi-sampling) is in some sense very similar to these two problem. It would be better if the authors includes the comparisons of the different goals.

**Minor Suggestions:**

1. In Algorithm 2, based on the output line it seems that condition $m<n$ is needed.

2. In Section 3, the DP mainlt comes from random shuffling. Why single sampling part uses pure DP yet multi-sampling part converts to approx DP? Is there a barrier

3. In first paragraph of Section 4.1.2, Algorithm 3 should be 4.

4. In Section 5.3. \|D,\hat{D}\| should be \|D - \hat{D}\| to maintain the notation in the paper.

**Paper Award:**

No

---

### Author Rebuttal · Authors · 2024-11-23

**Review 1**

The model is only based on the k-ary distributions and Gaussian distributions.

> **Author Response:** Although our results indeed focus on two classes of distributions, we note that our definition of DP multi-sampling accommodates arbitrary distributions.

I wonder if similar method is also applicable to other distributions.

> **Author Response:** The weak multisampling algorithm based on amplification by shuffling at its core relies on the existence of good local randomizers R such that on input $X \sim D$, $R(X)$ is simultaneously $\alpha$ close to D in TV distance as well as satisfies $\epsilon_0$-DP for $\epsilon_0 = O(\log n)$ where $n$ is the size of input dataset. In this paper, randomized response turned out to be a good local randomizer for k-ary distributions.
>
> We tried constructing local randomizers for other classes of distributions as well such as Gaussian distributions but those randomizers gave us suboptimal sample complexities. Making sure that $R(X)$ was $\alpha$ close to $D$ in TV distance led the sample complexities to increase by a lot.

Is the TV distance is the bottleneck?

> **Author Response:** One can definitely try to look at other distance metrics between distributions other than TV distance. One can also try to design new types of randomizers and hopefully they will work better for other distribution classes.

In Algorithm 2, based on the output line it seems that condition $m < n$ is needed.

> **Author Response:** Yes, that's correct.

In Section 3, the DP mainlt comes from random shuffling. Why single sampling part uses pure DP yet multi-sampling part converts to approx DP? Is there a barrier.

> **Author Response:** As mentioned in the paper, we use the amplification by shuffling result by Feldman et al that does this. In essence, the result says that if we have local randomizers that provide $\epsilon_0$ local DP for each input data point then randomly shuffling the results gives us $(\epsilon, \delta)$ DP. A better amplification by shuffling the result will make our result stronger.

In first paragraph of Section 4.1.2, Algorithm 3 should be 4.

> **Author Response:** Yes, noted

In Section 5.3. $|D,\hat{D}|$ should be $|D - \hat{D}|$ to maintain the notation in the paper.

> **Author Response:** Yes, noted

---

> ### Author Rebuttal · Authors · 2024-11-23
>
> There are lots of materials on private histogram algorithm, where the algorithm takes iid data as input and output a distribution with approximate histogram distribution. Take arxiv.org/abs/0811.2501 for example, which measures accuracy in L2 and Kolmogrov distance. There are also private synthetic data algorithms which takes an input dataset and output a new dataset with similar distribution, e.g. arxiv.org/abs/2302.05552. It seems that private sampling (especially weak multi-sampling) is in some sense very similar to these two problem. It would be better if the authors includes the comparisons of the different goals.
>
> > **Author Response:** Thanks for the suggestion. The reviewer is indeed correct, there is a series of private synthetic data algorithms. Private sampling is in a sense a similar problem. The work on DP (single- and multi-) sampling has two distinguishing features (1) we assume the sensitive data is drawn i.i.d. from some distribution [though our privacy holds for any input] (2) the objective is to minimize TV distance between our generated samples' distribution and the underlying distribution.
> >
> > The two papers that the reviewer cited each differ in at least one of these axes. In the paper "A Statistical Framework for Differential Privacy", the authors Wasserman and Zhou have a different way of measuring quality: with the Kolmogorov-Smirnov distance, they compare the CDF of the underlying distribution against an empirical CDF constructed from a DP algorithm's output.
> >
> > In the paper "Algorithmically Effective Differentially Private Synthetic Data", the error metric is the 1-Wasserstein distance between empirical measures. The authors He et al. do not assume the input is drawn i.i.d. from an underlying distribution like we do.
> >
> > We also note that we employ different techniques than the cited work (specifically, amplification-by-shuffling and Euclidean-Laplace noise).

---

### Author Rebuttal · Authors · 2024-11-23

**Review 2**

> **Author Response:** We thank the reviewer for the commentary. We acknowledge that the techniques used are quite simple but we contend this can be a boon for future implementation. For example, we anticipate that it will be quick to prototype ShuRR and evaluate on real-world datasets.

---

**Review 3**

There should be a statement of (informal) main theorem at the beginning of the paper, for the most important result

> **Author Response:** Thanks for the suggestion. Noted, we will try to add the main theorem at the beginning.

The lower bounds and upper bounds for the sample complexity does not match for discrete distributions.

> **Author Response:** Yes, it would have been ideal to have matching upper and lower bounds for the discrete case. However we wish to mention that the lower bound for strong multi sampling and the upper bound for weak multisampling does separate the two multi-sampling problems in certain regimes of m and k as mentioned in the last paragraph in section 5.1.

The Euclidean-Laplace mechanism for the gaussian distribution(equation (3)) does not seem to run in polynomial time.

> **Author Response:** In this paper, our primary focus was on optimizing the sample complexities while satisfying DP. However, in appendix E, we do provide an algorithm to samples from the Euclidean-Laplace distribution. Assuming that it takes O(1) time to generate a standard Gaussian sample and O(1) time to generate a sample from the standard exponential, the running time of our algorithm is O(d).
>
> In particular, to generate one random sample from ELap(b) we need one random sample from gamma distribution, Gamma(d,b) where d is the number of dimensions of the input dataset and d random samples from a Gaussian distribution N(0,1). Generating a sample from a Gamma(d,b) distribution can be done by adding d independent random samples from a Gamma(1,b) distribution (note that this only holds for integer d). A sample from a Gamma(1,b) distribution can be obtained by scaling a sample from Gamma(1,1) by multiplying it with b. Gamma(1,1) is the exponential distribution.

---

### Meta-Review · Area_Chair_Dqg1 · 2024-12-13

**Recommendation:** Accept
**Confidence:** 5

**Metareview:**

The paper extends the previously studied problem of privately generating a single sample given iid samples from a distribution into the problem of generating multiple samples, which brings multiple challenges. All of the reviewers agree that this direction is interesting and are encouraged by the positive results in the paper, including a general transformation from single-sample algorithms to multi-sample algorithms, and improved algorithms for specific families of distributions such as Gaussians. Overall, given the positive reviews, I recommend to accept the paper.

**Paper Award:**

No